# Enyne diketones as substrate in asymmetric Nazarov cyclization for construction of chiral allene cyclopentenones

Shengbiao Tang [1✉], Peng Zhang[1], Ying Shao[1] & Jiangtao Sun [1✉]

The Nazarov cyclization is one of the most powerful tools for the stereoselective synthesis of various cyclopentenone scaffolds. Therefore, developing the new classes substrate of Nazarov reaction is an important endeavor in synthetic chemistry. Herein, we report enyne diketones, enables diastereo- and enantioselective construction of chiral allene cyclopentenones in moderate to good yields with good enantioselectivities (up to 97% ee). Importantly, it is a typical example for asymmetric synthesis of cyclopentanones with allene moiety using Nazarov cyclisation. Mechanistic studies indicate that this metal-organo relay catalysis protocol involves a rhodium-catalyzed tandem oxonium ylide formation/[2,3]-sigmatropic rearrangement/reverse benzylic acid rearrangement, followed by organo-catalyzed asymmetric Nazarov cyclization/alkyne-to-allene isomerization to give the final chiral allene cyclopentenones.

---

[1] Jiangsu Key Laboratory of Advanced Catalytic Materials & Technology, School of Petrochemical Engineering, Changzhou University, 1 Gehu Road, 213164 Changzhou, China. ✉email: shengbiaotang@cczu.edu.cn; jtsun@cczu.edu.cn

Enantiopure cyclopentenones are significant structural motifs that are widely present in numerous natural products and bio-significant molecules[1–4]. For example, some representative chiral cyclopentenones bearing 5-hydroxy-quaternary stereocenter scaffolds are illustrated in Fig. 1a[5–10]. Nazarov cyclization, one of the few 4π electrocyclic reactions that can be controlled in an asymmetric fashion by chiral catalysts, is considered to be one of the most direct and convenient methods to these important compounds in recent years[1–4,11–14]. Therefore, extensive efforts have been devoted to catalytic asymmetric versions of the Nazarov cyclizations. Scince Trauner's pioneering the first asymmetric Nazarov cyclization in 2003[15], divinyl ketones became the most active research substrates for chemists over the past decades (Fig. 1b). The contribution works for asymmetric Nazarov cyclization of divinyl ketones includes the groups of Trauner[15,16], Aggarwal[17], Frontier[18], Rueping[19,20], Togni[21], Tius[22,23], Rawal[24], Tang[25,26], Zhou[27], List[28], Zhu[29] and so on

(Fig. 1b). In contrast, the other types substrate of asymmetric Nazarov cyclization are rarely reported. In 2010, Tius and co-workers designed a vinyl diketoesters substrate, which undergoes asymmetric Nazarov reactions to provide chiral cyclopentenones catalyzed by primary amine–thiourea catalyst (Fig. 1c)[30]. Later, the same group also reported Pd-catalyzed enantioselectve Nazarov cyclization of vinyl diketoesters substrate[31]. Furthermore, the group of Frontier developed an impressive dienyl diketones substrate that has been successfully used in the Nazarov reaction (Fig. 1d), however, most of their works were focused on the recemic electrocyclic reaction, few examples involving aysmmetric versions[32–35].

Despite the above progress have been made, the substrate classes in asymmetric Nazarov cyclization remains quite limited. As a results, the diversity of the chiral cyclopentenones that can not be obtained. For example, the chiral cyclopentenone which bearing allene moiety can not be prepared by the reported

**a** Representative examples of naturally occurring cyclopentenones with a 5-hydroxy-bearing quaternary stereocenter

(-)-tylopilusin B    cucurbitarins D    nvolutenone A (R = Me) involutenone B (R = CH$_2$OH)    przewalskin

curcusone D    2-*epi*-hydroxyisojatrogrossidion    hitoyol B    pentenomycin

**b** Divinyl ketones: dominant substrates in asymmetric Nazarov cyclization (well-established)

divinyl ketones    chiral conditions / Includes: BA, LA, [Cu] [Ni], [Fe], [Sc], [Gr], [Co]    4π electrocyclization

**c** Vinyl diketoesters: developed by Tius (2010)

vinyl diketoesters    bifunctional organocatalytic    4π electrocyclization

**d** Dienyl diketones: developed by Frontier (only 7 examples shown in asymmetric version)

divinyl diketones    1,6-conjugate addition initiated NuH base    4π electrocyclization

**Fig. 1 Selected naturally occurring chiral cyclopentenones, the known substrates in asymmetric Nazarov cyclization for construction of chiral cyclopentenone. a** Representative examples of naturally occurring cyclopentenones with a 5-hydroxy-bearing quaternary stereocenter. **b** Divinyl ketones: dominant substrates in asymmetric Nazarov cyclization. **c** Vinyl diketoesters: developed by the group of Tius. **d** Dienyl diketones: developed by the group of Frontier.

Nazarov cyclization yet. In this regard, developing new classes substrate and catalytic strategies for asymmetric Nazarov cyclization to construction novel chiral cyclopentenones are highly desirable.

Herein, we report a metal-organo relay catalysis approach for the construction of chiral allene cyclopentenones in high diastereo- and enantioselectivities from 1,4-enyne alcohols with diazo compounds (Fig. 2a). The intermediate enyne diketones play a key role in the process of asymmetric Nazarov cyclization to form target compounds.

Based on literature reports[30–36] and the control experiments and mechanism studies, the possible reaction mechanism of our design has been proposed (Fig. 2b). The formation of chiral allene cyclopentenone might proceed via a sequential process.

First, the reaction between rhodium catalyst and diazoacetate **1** generates rhodium carbene species **Int-1**. Nucleophilic addition of **2** to **Int-1** affords ylide intermediate **Int-2**, which undergoes a [2,3]-sigmatropic rearrangement yields **Int-3**. This intermediate would form 1,2-diketone-enyne **3** via reverse benzilic acid rearrangement[37,38]. Exposing **3** to a chiral thiourea catalyst may generate intermediate **Int-4** via hydrogen-bonding and keto-enolic tautomerism. Intramolecular Nazarov cyclization delivers the cyclopentenone alkyne intermediate **Int-5** containing two adjacent stereogenic centers. Under such reaction conditions, alkyne-to-allene isomerization occurs and undergoes central-to-axial chirality transfer[39–41] to give the final product (−)-**4**.

**Fig. 2 Our discovery and proposed mechanism. a** This work: enyne diketones: a design substrate in aysmmetric Nazarov cyclization. **b** Proposed catalytic cycle of our design.

## Results

**Reaction optimization.** Inspired by the seminal reports by Davies[36,42] and in continuation with our research interest in metal-carbene chemistry[43–47]. We conducted the reaction of diazoacetate **1a** with 1,4-enyne-3-ol **2a** bearing both an allylic and a propargylic moiety in the presence of Rh$_2$(OAc)$_4$, an enyne diketone **3** was generated in 92% NMR yield, which indicated that a reverse benzylic acid rearrangement of **Int-3** occurred under such reaction conditions (Fig. 2b)[37,38]. Furthermore, subjecting enyne diketone **3** to silica gel, the cyclopentenone **4** was isolated in nearly 10% yield, this result revealed that the Nazarov cyclization was occurred. These unexpected experimental results and unconventional allene cyclopentenone structure prompted us to further explore the reaction.

We began to thoroughly investigate the reaction. With a view to establishing the optimal reaction conditions, phenyl diazoacetate **1a** and 1,4-enyne alcohol **2a** were used as model substrates (Table 1). A variety of diazoesters **1b–1f** instead of **1a** were tested under Method A, and no better results obtained (Table 1, entries 2-6). Intensively screening of the reaction conditions revealed that, using [Rh$_2$(OAc)$_4$] (2 mol%) as the catalyst and hexane as the solvent (rt for 12 h), with addition of MgO and CH$_2$Cl$_2$, allene cyclopentenone **4** was obtained in 79% yield in single isomer (Method A, one-pot process, see Table 1 and Supplementary Table 1 for details). Screening of the solvents revealed that hexane was still the best one (Table 1, entries 1, 7 and 8). Further evaluation of other rhodium catalysts, such as Rh$_2$(TFA)$_4$ and Rh$_2$(esp)$_4$ gave inferior results (entries 9-10). When SiO$_2$ was used instead of MgO, only 43% yield of **4** was obtained (entry 11).

**Substrates scope of racemic version.** Having established the optimal reaction conditions for diastereoselective synthesis (Method A), we next started to investigate the scope of reactions (Fig. 3, all examples dr >19:1). The scope of enyne alcohols **2** was first evaluated. Generally, the R$^3$ moiety of aryl-1,4-enyne alcohols containing either an electron-donating or an electron-withdrawing group at the *para-* or *ortho-*position of the phenyl ring were well tolerated, giving the corresponding products (**4-9**) in 63-82% yields with excellent diastereoselectivity (>19:1). However, the yield decreased when phenyl ring bearing

4-methoxycarbonyl group **10**. Reactions of 4-Ph-phenyl, 3-cholor-phenyl, 2-cholorphenyl and 2-naphyl substituted enyne alcohols with **1a** proceeded smoothly to give the desired products (**11–14**) in good yields, with high diastereoselectivity. The substrates with alkylated R$^3$ group such as *n*-butyl, 3-Cl-proparyl, cyclopropyl and *tert*-butyl was also compatible, providing moderate yields of the corresponding products **15–18** (66–72%, >19:1 dr). Next, different R$^1$ and R$^2$ substituents were also explored, delivering **19–23** in 61–83% yields with >19:1 dr values. Obviously, the increased steric hindrance resulted in decreased yields.

Next, we further assessed the generality of diazo compounds **1**. With respect to the aryl diazoacetates, both electron-donating groups and electron-withdrawing groups at the *para*-position of phenyl rings proceeded smoothly to deliver the desired products **24–26** and **28–31** in moderate to good yields. The use of 4-BocNH substituted phenyldiazoacetate gave **27** in 38% yield. Aryldiazoacetates bearing an electron-donating (-Me and -OMe) or electron-withdrawing group (-Cl) at the *mata*-position of the phenyl ring reacted well, affording the corresponding products **32–34** in 68–74% yields. However, the use of *ortho*-methyl-substituted phenyldiazoacetate only afforded the desired products **35** in 34% yield. Furthermore, the *di-* and *tri*-substituted phenyl diazoacetates were also tested, providing the desired products **36–39** in moderate to good yields, with execellent dr values. Moreover, 2-naphthyl and heterocycles, such as 1,3-benzodioxole, Boc-indole ring and 3-thiophene were all tolerated, affording the desired products **40–43** in good yields. However, pyridine diazoacetate also work and furnishing allene cyclopentenone **44** in 36% yield.

**Optimization of enantioselective synthesis.** With the successful distereoselective synthesis of allene cyclopentenone **4**, we then wish to realize the enantioselective synthesis of chiral allene cyclopentenone **4** by establishing a highly effective asymmetric catalytic system. Intensive attempts on transition-metal catalysis failed to reach high enantioselectivity. We then turned out our attention to use organo catalysts. However, preliminary evaluation of chiral phosphoric acids led to recover the starting materials (see Supplementary Table 2 for details). Next, cinchona-

---

**Table 1 Selected optimization[a].**

| entry | diazo | variation from conditions A | 4 (yield, %)[b] |
|---|---|---|---|
| 1 | **1a** | none | 79 |
| 2 | **1b** | none | 68 |
| 3 | **1c** | none | 55 |
| 4 | **1d** | none | 64 |
| 5 | **1e** | none | 25 |
| 6 | **1f** | none | 52 |
| 7 | **1a** | toluene instead of hexane | 54 |
| 8 | **1a** | CH$_2$Cl$_2$ instead of hexane | 31 |
| 9 | **1a** | Rh$_2$(esp)$_4$ instead of Rh$_2$(OAc)$_4$ | 65 |
| 10 | **1a** | Rh$_2$(TFA)$_4$ instead of Rh$_2$(OAc)$_4$ | 63 |
| 11 | **1a** | SiO$_2$ instead of MgO, 24 h | 43 |

[a]Method A: **1** (0.3 mmol, 1.5 eq.), **2a** (0.2 mmol, 1eq.) and Rh$_2$(OAc)$_4$ (2 mol%) in hexane (2 mL) stirred at rt. for 12 h under N$_2$. Then, to the solution were added MgO (200 mg, 50 eq.), CH$_2$Cl$_2$ (5 mL) and stirred at 40 °C for another 3 h. [b]Isolated yield, dr > 19:1 (determined by $^1$H NMR analysis).

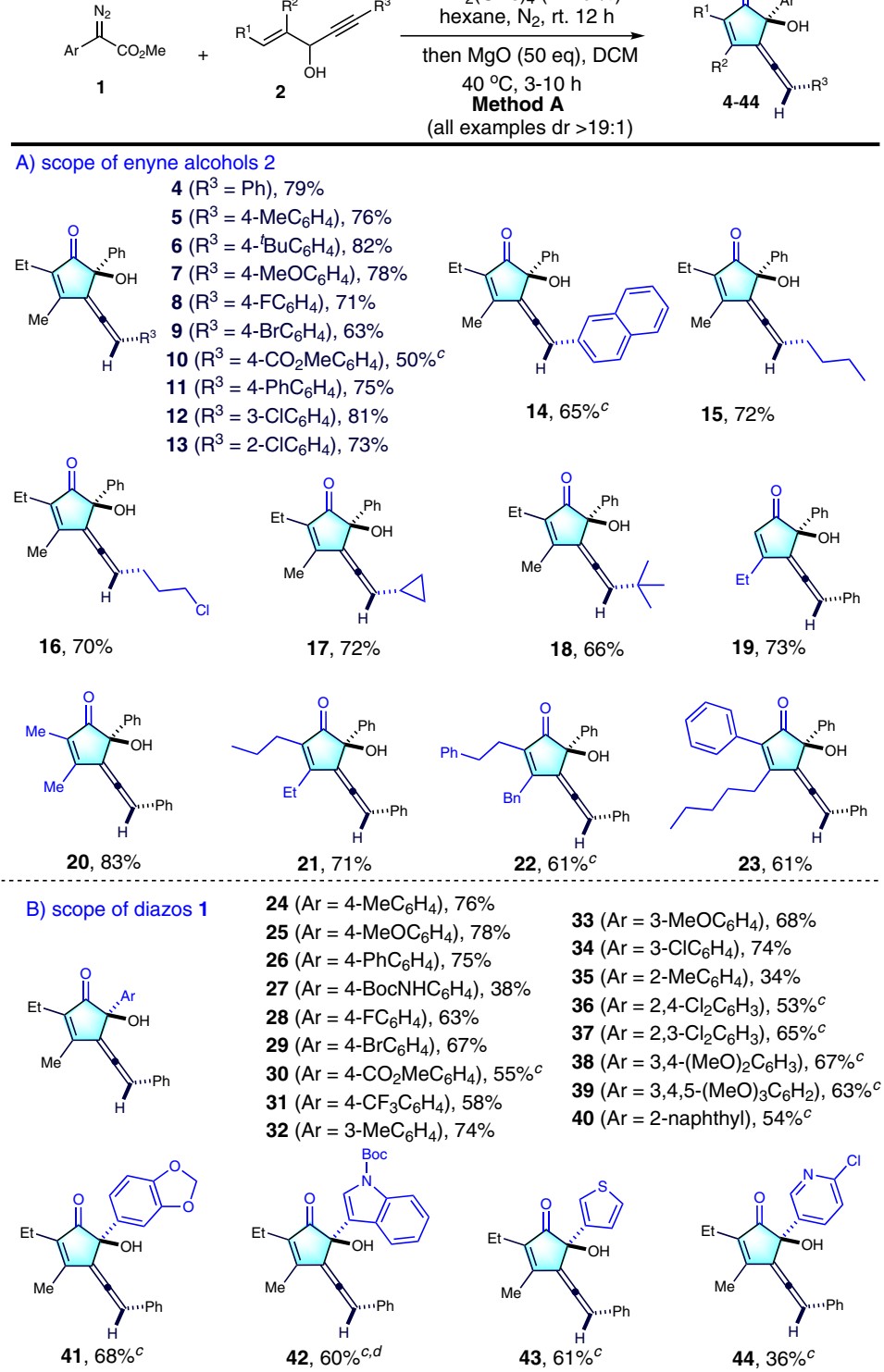

**Fig. 3 Racemic version scope of substrates.** [a]Method A: **1** (0.3 mmol, 1.5 eq.), **2** (0.2 mmol, 1eq.) and Rh$_2$(OAc)$_4$ (2 mol%) in hexane (2 mL) stirred at rt. for 12 h under N$_2$. Then, to the solution were added MgO (200 mg, 50 eq.), CH$_2$Cl$_2$ (5 mL) and stirred at 40 °C for another 3 h. [b]Isolated yield, dr > 19:1 (determined by $^1$H NMR analysis). [c]Sovent of first step: (toluene:hexane = 1:1, 2 mL) instead of hexane. [d]**1** (2.0 eq.) was used.

based chiral bases were examined (Fig. 4). Further studies revealed that the use of quinidine **Q1** gave **4** in 76% yield but almost no enantioselectivity, whereas **Q2** led to poor yield and low ee. The use of bifunctional chiral squaramides **Q3**-**Q4** showed low catalytic reactivity, affording both low yields and low ee values. Subsequent utilization of amine-thiourea catalyst **Q5**–**Q6** gave promising results,

leading to moderate yield with >60% enantioselectivity. To further improve the enantioselectivity, we synthesized trifunctional binaphthyl diamine catalyst **Q7**[48] and used it in this reaction. The initial test delivered **4** in 63% yield and 82% ee. Suprisingly, when we prepared a large amount of **Q7** with a view to screening other reaction conditions, only trace amount of **4** with 52% ee was obtained. After careful analysis, we found that we actually used **Q9**

**Fig. 4 Screening the chiral organo-catalysts.** Conditions: **1a** (0.3 mmol, 1.5 eq.), **2a** (0.2 mmol, 1eq.) and $Rh_2(OAc)_4$ (2 mol%) in hexane (2 mL) stirred at rt. for 12 h under $N_2$; Then, removing the solvent and DCM (2.0 mL), **Q9** (10 mol%) were added. The mixture was stirred for another 72 h. Isolated yield for two steps. Ee values were determined using chiral HPLC.

Further screening the solvents indicated that cyclopentyl methyl ether (CPME) was the best one (Table 2, entries 1-7, and see Supplementary Table 3 for details). In screening additives, we found that 4 Å MS could improve the enantioselectivity (entry 7–8). The examination of organo-catalyst loading shown that 15 mol% gave a better yield (entries 9–11). As a result, the optimal reaction conditions for asymmetric catalysis had been established (**Method B**): **1a** (0.3 mmol, 1.5 eq.), **2a** (0.2 mmol, 1eq.), and $Rh_2(OAc)_4$ (2 mol%) in hexane (2 mL) stirred at room temperature for 12 h. Then, removing the solvent and CPME (2.0 mL), **Q9** (15 mol%) and 4 Å MS (40 mg) were added. The mixture was stirred for another 72 h.

**Substrate scope**. Having established the optimal reaction conditions for enantioselective synthesis (Method B), we next started to investigate the scope of reactions (Fig. 5). The scope of enyne alcohols **2** was first evaluated. Generally, the $R^3$ moiety of aryl-1,4-enyne alcohols containing either an electron-donating or an electron- withdrawing group at the *para*- or *ortho*-position of the phenyl ring were well tolerated, giving the corresponding chiral products (–)-(**4**) to (–)-**10** in 48-66% yields and with 87-97% ee. Reactions of 4-Ph-phenyl, 3-cholor-phenyl, 2-cholorphenyl and 2-naphyl sub-stituted enyne alcohols with **1a** proceeded smoothly to give the desired chiral products (–)-(**11**) to (–)-**14** in good yields, and high ee values. The substrates with alkylated $R^3$ group such as *n*-butyl, 3-Cl-proparyl, cyclopropyl and *tert*-butyl were also compatible, providing moderate yields of the corresponding products (–)-(**15**)-(–)**18** (50-64%, 86-89% ee, >19:1 dr). Next, different $R^1$ and $R^2$ substituents were also explored, delivering (–)-(**19**) to (–)-**23** in 47-76% yields with 81-97% ee. Obviously, the increased steric hindrance resulted in decreased yields and ee values. Next, we further assessed the generality of diazo compounds **1** (Fig. 5). With respect to the aryl

## Table 2 Selected optimization of enantioselective synthesis[a].

| entry | solvent | cat. (x mol%) | 4 (yield, %)[b] | ee (%)[c] |
|---|---|---|---|---|
| 1 | $CH_2Cl_2$ | **Q9** (10) | 63 | 82 |
| 2 | DCE | **Q9** (10) | 52 | 76 |
| 3 | toluene | **Q9** (10) | 30 | 66 |
| 4 | $Et_2O$ | **Q9** (10) | 64 | 89 |
| 5 | hexane | **Q9** (10) | 18 | 86 |
| 6 | $^tBuOMe$ | **Q9** (10) | 59 | 83 |
| 7 | CPME | **Q9** (10) | 62 | 91 |
| 8[d] | CPME | **Q9** (10) | 61 | 95 |
| 9[d] | CPME | **Q9** (5) | 36 | 94 |
| 10[d] | CPME | **Q9** (15) | 65 | 95 |
| 11[d] | CPME | **Q9** (20) | 65 | 95 |

[a]Conditions: **1** (0.3 mmol, 1.5 eq.), **2a** (0.2 mmol, 1eq.) and $Rh_2(OAc)_4$ (2 mol%) in hexane (2 mL) stirred at rt. for 12 h under $N_2$; Then, removing the solvent and CPME (2.0 mL), **cat.** were added. The mixture was stirred for another 72 h; [b]Isolated yield for two steps; [c]Ee values were determined using chiral HPLC; [d]4 Å MS (40 mg) was uesd as additive. CPME: cyclopentyl methyl ether.

not **Q7** in the first try because **Q7** can be easily converted to **Q9** at higher temperature during the preparation process. Likewise, cinchonidine derived thiourea **Q8** afforded **4** in 66% yield with 78% ee, whereas hydroquinidine derived **Q10** gave 46% yield of **4** with 72% ee. In addition, the catalyst **Q11**–**Q20** were also examined, unfortunately, there are no better results were obtained (see Supplementary Table 2 for details). Given **Q9** exhibited the best catalytic activity in this reaction, other factors were further investigated, and as shown in Table 2.

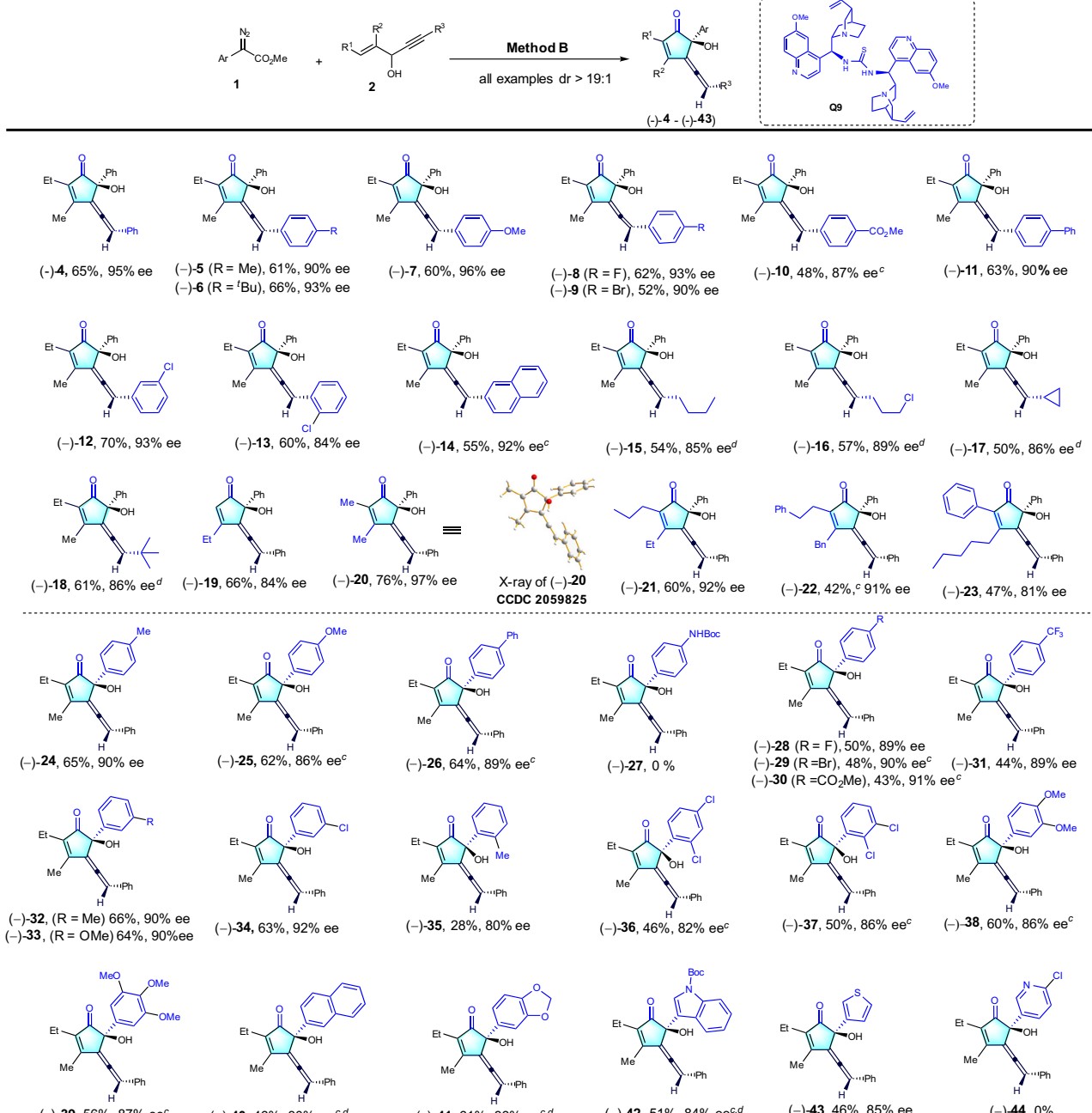

**Fig. 5 Enantioselective version substrate scope.** [a]Method B: **1** (0.3 mmol, 1.5 eq.), **2** (0.2 mmol, 1eq.), and Rh$_2$(OAc)$_4$ (2 mol%) in hexane (2 mL) stirred at room temperature for 12 h. Then, removing the solvent and CPME (2.0 mL), **Q9** (15 mol%) and 4 Å MS (40 mg) were added. The mixture was stirred for another 72 h. [b]Isolated yields; all examples dr >19:1; the ee values were determined by chiral HPLC analysis. [c]Mixed solvent (toluene/hexane = 1:3, 2 mL) instead of hexane in first step. [d]The second step of method B for 96 h, and then the MgO (400 mg) was added and stirred for another 5 h.

diazoacetates, both electron-donating groups and electron-withdrawing groups at the *para*-position of phenyl rings proceeded smoothly to deliver the desired products (-)-**24**-(-)-**31**) in moderate to good yields with 86-91% ee for the corresponding chiral allenes. Aryldiazoacetates bearing an electron-donating (-Me and -OMe) or electron-withdrawing group (-Cl) at the *mata*-position of the phenyl ring reacted well, affording the corresponding products (−)-(**32**)-(−)-**34** in 63-66% yields with 90-92% ee. However, the use of *ortho*-methyl-substituted phenyldiazoacetate only afforded the desired products (−)-**35** in 28% yield with 80% ee, which probably attributed to the large steric hindrance of methyl group. Furthermore, the *di*- and *tri*-substituted phenyl diazoacetates were also tested, providing the desired products (−)-**36**-(−)-**39** in moderate to

good yields, with good ee values. Moreover, 2-naphthyl and heterocycles, such as 1,3-benzodioxole, Boc-indole ring and 3-thiophene were all tolerated, affording the desired products (−)-**40**-(−)-**43** in good yields and good enantioselectivities (84-86% ee). The absolute configurations in these chiral allenes were established by X-ray single-crystal diffraction study[49]. It should be noted that the product (-)-**27** and (-)**44** were not obtained in the optimal asymmetric conditions (Method B).

**Control experiments and mechanism studies.** To gain insight into the reaction mechanism, control experiments were performed (Fig. 6). First, the reactions of **1a** with **2 v** and **2w** under

**Fig. 6 Control experiments and mechanism studies. a** 1,4-enyne allylic alcohol as substrate. **b** 1,4-dienyl allylic alcohol as substrate. **c** monoallylic alcohol as substrate. **d** Deuterium experiment. **e** The coversation of compound **48** in different time periods. **f** Both **Q9** and SiO2 accelerate (−)-36′ conversion to allene (−)-36. **g** (−)-36′ in the condition of base or CH₂Cl₂.

rhodium catalysis furnished the corresponding 1,2-diketones **45** and **46** in 71% and 58% yields, respectively (Fig. 6a, b). However, the use of **2x** as substrate to reaction with **1a**, as we expected, no corresponding diketone **47** was observed, instead, the O-H insertion product **47'** was obtained in 68% yield (Fig. 6c). These results probably suggested that the π-σ(C-OH)-π is essential structure of alcohol to occur the sequential ylide formation/[2,3]-sigmatropic rearrangement and reverse benzylic acid rearrangement to form the 1,2-diketone. Next, we conducted the deuterium experiment for further insight into mechanism process. The addition of D₂O to the reaction of **1a** and **2a** in the conditions of method B and method A, gave **D-4** with 70% and 30% deuterium labeling, respectively (Fig. 6d). The reason of **Q9** give the better deuterium labeling ratio than MgO probably due to the **Q9** has higher ability to stable intermediate anion, and the result also supported that the proton transfer has occurred in the path from alkyne to allene. It is worth noting that both alkyne-containing

alcohol (−)-**36′** and allenol (−)-**36** were formed during the enantioselective cyclo-isomerization of diketone **48** (Scheme 6e). The yields of (−)-**36′** and (−)-**36** were time-depending. The amount of (−)-**36** increased by prolonging the reaction to 92 h. In contrast, the yield of (−)-**36′** increased to 35% within 48 h and then decreased with longer reaction time (Fig. 6e). To further validate the prerequisite formation of (−)-**36′**, subjecting (−)-**36′** to SiO₂ or organic catalyst **Q9**, both of them are led to high yield of (−)-**36** with enantioselectivity retention (Fig. 6f). Indeed, in the presence of base, such as Et₃N, the isomerization from alkyne-to-allene also proceeded smoothly. Even absence of catalyst, the isomerization of (−)-**36′** also could occur when stirred in the solvent of dichloromethane (Fig. 6g).

**Gram scale and synthetic application.** Further derivatization reactions were performed to demonstrate the promising synthetic

**Fig. 7 Gram scale and synthetic application. a** Gram scale and Reduction reaction. **b** Tandem cyclization reaction induced by TfOH.

practicality of this method (Fig. 7). Treatment of **1a** with **2b**, followed by addition of 1,2-diaminobenzene gave heterocycle **49** in 89% yield (see Supplementary for details). The structure of **49** was determined by single-crystal X-ray diffraction, which also confirmed the structures of enyne diketone intermediates. To further assess the utility of this tandem annulation, a gram scale synthesis was conducted, affording (−)-**20** in 68% yield with no significant ee erosion (1.112 g) (Fig. 7a). Then, conversion reactions of (−)-**20** were conducted. Subjecting (−)-**20** with LiAlH$_4$ yielded 1,2-diol **50** in 92% yield with 94% ee. Addition of methyl Grignard reagent to (−)-**20** led to tertiary alcohol **51** in a good yield with 95% ee. Finally, selective hydrogenation of allene moiety provided **52** in 96% yield and 96% ee. Interestingly, the reaction of **20** with TfOH in dichloromethane, a racemic condensed polycyclic compound **53** was generated (Fig. 7b).

## Discussion

In this work, we describe an efficient protocol to construct chiral allene cyclopentenones from 1,4-enyne alcohols with diazo compounds via metal-organo relay catalysis strategy. The intermediate enyne diketones represent an underdevelop classes of substate in asymmetric Nazarov cyclization.

## Methods

**General procedure for the synthesis and experiment data of (±)-4-44 and (-)-4-(-)43**. Method A (one-pot process for racemic products): To the tube was added freshly distilled hexane (2.0 mL), diazo **1** (0.3 mmol), enyn-3-ol **2** (0.2 mmol) and then Rh$_2$(OAc)$_4$ (2.6 mg, 2 mol%) was added to the mixture solution. The tube was sealed and stirred at rt for 12 h. And then MgO (400 mg, 50 eq.) and CH$_2$Cl$_2$ (5 mL) was added to the mixture solution, and stirred at 40 °C for another 3 h. Then the mixture solution was filtrated by diatomite, and wash with EtOAc (10*3). The filtrate was concentrated by rotary evaporation. The crude product was

purified by silica gel column chromatography corresponding eluent to afford the desired products (±)-**4-44**.

**Method B** (for asymmetric version): To the dried tube was added freshly distilled hexane (2.0 mL), diazo **1** (0.3 mmol), enyn-3-ol **2** (0.2 mmol), and then Rh$_2$(OAc)$_4$ (2.6 mg, 2 mol%) was added to the mixture solution. The tube was sealed and stirred at rt. for 12 h. The solvent was removed directly; and then, the CPME (2.0 mL), **Q9** (15 mol%) and 4ÅMs (40 mg) was added to the tube, stirred at rt. for another 72 h. The solution was concentrated under reduced pressure and purified by flash chromatography corresponding eluent to afford the desired products (-)-**4-43**.

## Data availability

All data generated in this study are provided in the Airticle and Supplementary Information as well as from the corresponding author on request. The X-ray crystallographic data used in this study have been deposited at the Cambridge Crystallographic Data Centre (CCDC) with the accession code CCDC 2059825 for (−)-**20** and CCDC 2059828 for **49**, CCDC 2133232 for **53**. These data can be obtained free of charge from The Cambridge Crystallographic Data Centre via www.ccdc.cam.ac.uk/data_request/cif.

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

## Acknowledgements
We thank the National Natural Science Foundation of China (21901022, 21971026, 22171028) and the Jiangsu Key Laboratory of Advanced Catalytic Materials and Technology (BM2012110) for their financial support.

## Author contributions
S.T. conceived the idea, performed the most experiments and wrote the draft manuscript; P.Z. assisted in some experiments and helped in synthesis of substrates. Y.S. collected and analyzed the data; J.S. guided this project and revised the manuscript.

## Competing interests
The authors declare no competing interests.
