## [Peer Review File · Nature Communications]

REVIEWER COMMENTS

Reviewer #1 (Remarks to the Author):

This manuscript described a novel organo-catalyzed asymmetric Nazarov cyclization of enyne diketones which were prepared via a rhodium-catalyzed tandem oxonium ylide formation/[2,3]-sigmatropic rearrangement/reverse benzylic acid rearrangement. By using 1,4-enyne-3-ols bearing both an allylic and a propargylic moiety as starting substrates, a series of novel enyne diketones followed by cyclopentanones with allene moiety have been achieved with broad substrate scope, moderate to good yields with good enantioselectivities under mild reaction conditions. Furthermore, a reasonable and innovative mechanism has been proposed, new compounds seem to be adequately characterized and experimental procedures are, for the most part, clear and complete. Given the above, I am pleased to recommend this article to be accepted for publication in Nature Communications.

Note:

1. Ref. 42 need to added title of the article.
2. In SI, Table S3, orgnocatalyst "IX" should be "Q9".

Reviewer #2 (Remarks to the Author):

This manuscript from Sun and Tang described an interesting one-pot sequential construction of chiral allene cyclopentenones by a relay-catalysis of rhodium (II) and chiral thiourea catalysts. This method underwent very elaborate process, including O-H insertion of rhodium carbene, [2,3]-sigmatropic rearrangement, retro-benzylic acid rearrangement, asymmetric Nazarov reaction, and isomerization of alkynes. And what's incredible is the basically very good yields with good to excellent enantioselectivities they have achieved in such a complicated reaction sequence. The broad substrate scope demonstrated the synthetic potential of this method. This work also provides new type of substrates for Nazarov Cyclization. In view of these merits, I think this manuscript could be published after addressing and correcting the followings:

1. Page 2, 2nd paragraph, "Lewis catalyzed". Page 2, line 10, right column, "Lewis SiO₂". These expressions are confusing.
2. The authors proposed the formation of 1,2-diketone-enyne 3 via the retro-benzilic acid rearrangement of the [2,3]-sigmatropic rearrangement product Int-3. They have detected and isolated the 1,2-diketone-enyne product in the rhodium catalyzed reaction of diazo compound and 1,4-enyne alcohol, but no int-3 has been detected yet. Is it because the int-3 is very unstable or is it ever formed during the generation of the 1,2-diketone-enyne 3? I suggest the author should figure this out by experiments.
3. Scheme 2, in the graph, scope A, the Ar should be R³? It is very confusing.
4. Page 4, line 7, left column, "4ÅMS" a space was missing.
5. Scheme 4c, the control experiment on reaction between 1a and 2x was carried out, but no diketone product was observed. I wondered what products they have detected, any [2,3]-sigmatropic rearrangement product? or O-H insertion product? or something else? Maybe this information could help the author further understand the mechanism of the current reaction.
6. Scheme 4f, they found both SiO₂ and the chiral thiourea Q9 could accelerate the isomerization of alkynes. The base could promote this process is reasonable, since MgO they used in the reaction worked well, Q9 bearing two tertiary amine functional groups could play as the base. How to explain the SiO₂ works as well?
7. I suggest moving the scheme 5a to the SI file, since this transformation may weaken the theme.
8. Can they make a gram scale synthesis to illustrate the reaction practicability?
9. The "Discussion" section should be conclusion.
10. The format of the references are not unified.

Reviewer #3 (Remarks to the Author):

This manuscript describes the discovery of enyne diketones as new substrates for Nazarov cyclization to construct chiral cyclopentenones bearing allene functionality. The reaction involves multi-step processes including rhodium-catalyzed tandem oxonium ylide formation/[2,3]-sigmatropic rearrangement, reverse benzilic acid rearrangement, organo-catalyzed Nazarov cyclization, and alkyne-to-allene isomerization. The results are interesting. However, the key chemistry of rhodium-catalyzed reaction of allylic as well as propargylic alcohols with diazoacetates via tandem oxonium ylide formation followed by [2,3]-sigmatropic rearrangement (Davies, 2010, 2012), and the Nazarov cyclization of dienyl diketones (Frontier, 2011-2015) has been known. The

new aspect is that enyne diketone intermediates obtained by rhodium catalysis was also found to undergo a similar Nazarov cyclization. Thus, I think the conceptual novelty of this manuscript is still limited due to the work by Davies and Frontier. Moreover, mechanistic novelty of stereochemical induction is also an important factor. There is no discussion on what is the rationale of how the thiourea catalysts helped to control the stereoselectivity and what is the mechanistic insight for stereospecific alkyne-to-allene isomerization. On the other hand, this manuscript is not well-organized and English has many errors and grammatical problems that need improvement. Overall, in this reviewer's opinion, this work is likely more suitable for publication in a less competitive journal such as Chem. Sci. after proper alteration.

1. What does the dr refer to? Please clearly indicate the structure of the minor isomer. The details for determination of the dr ratio by crude NMR should be included in the Supplementary Material (SI).
2. Please clearly indicate whether the formation of alkyne-containing alcohol product was observed. It would be of great interest if both alkyne-containing and allene-containing cyclopentenones could be obtained under proper conditions.
3. The preparation and characterization data of thiourea catalysts should be provided in SI.
4. In most cases, 1,4-enyne alcohols with R1 as ethyl and R2 as methyl were employed.
5. It seems not clear why the addition of D₂O gave D-4 with only 30% deuterium labeling (Scheme 4d).
6. What is the mechanistic model for the retained enantioselectivity in the alkyne-to-allene isomerization process under Lewis acidic SiO₂ and thiourea catalyst (Scheme 4f)?
7. Please provide a reaction pathway to illustrate the formation of product 53 in Scheme 5c.

Dear reviewers,

Enclosed please find our revised manuscript entitled “Enyne Diketones: A New Classes Substrate in Asymmetric Nazarov Cyclization for Construction of Chiral Allene Cyclopentenones” that we are submitting for publication in *Nature Communications*. We are grateful for your careful analysis of our manuscript, and we hope you will be convinced by our revised manuscript. Below please find the copies of the reviewer’s comments and our point-by-point response.

Responds to the reviewer #1’s comments:

(1) **Comments:** “This manuscript described a novel organo-catalyzed asymmetric Nazarov cyclization of enyne diketones which were prepared via a rhodium-catalyzed tandem oxonium ylide formation/[2,3]-sigmatropic rearrangement/reverse benzylic acid rearrangement. By using 1,4-enyne-3-ols bearing both an allylic and a propargylic moiety as starting substrates, a series of novel enyne diketones followed by cyclopentanones with allene moiety have been achieved with broad substrate scope, moderate to good yields with good enantioselectivities under mild reaction conditions. Furthermore, a reasonable and innovative mechanism has been proposed, new compounds seem to be adequately characterized and experimental procedures are, for the most part, clear and complete. Given the above, I am pleased to recommend this article to be accepted for publication in *Nature Communications*.”

Our reply: Many thanks for these positive comments. Your kind suggestions also help us a great in improving our manuscript.

(2) **Comments:**” Note: 1. Ref. 42 need to added title of the article. 2. In SI, Table S3, orgnocatalyst “IX” should be “Q9”.

Our reply: We appreciate these suggestions. The corrections have been made in the revised version.

Responds to the reviewer #2’s comments:

(1) **Comments:**” This manuscript from Sun and Tang described an interesting one-pot sequential construction of chiral allene cyclopentenones by a relay-catalysis of rhodium (II) and chiral thiourea catalysts. This method underwent very elaborate precess, including O-H insertion of rhodium carbene, [2,3]-sigmatropic rearrangement, retro-benzilic acid rearrangement, asymmetric Nazarov reaction, and isomerization of alkynes. And what’s incredible is the basically very good yields with good to excellent enantioselectivities they have achieved in such a complicated reaction sequence. The broad substrate scope demenstrated the synthetic potential of this method. This work

also provides new type of substrates for Nazarov Cyclization. In view of these merits, I think this manuscript could be published after addressing and correcting the followings”.

Our reply: We appreciate your positive comments. Please see below for our detailed responses.

(2) **Comments:** “1. Page 2, 2nd paragraph, “Lewis catalyzed”. Page 2, line 10, right column, “Lewis SiO2”. These expression are confusing.”

Our reply: Correction has been made in the revised version.

(3) **Comments:** “2. The authors proposed the formation of 1,2-diketone-enyne 3 via the retro-benzilic acid rearrangement of the [2,3]-sigmatropic rearrangement product Int-3. They have detected and isolated the 1,2-diketone-enyne product in the rhodium catalyzed reaction of diazo compound and 1,4-enyne alcohol, but no int-3 has been detected yet. Is it because the int-3 is very unstable or is it ever formed during the generation of the 1,2-diketone-enyne 3? I suggest the author should figure this out by experiments.”

Our reply:

Thanks for this comment. As requested, we conducted the experiment again. Unfortunately, it has not been observed yet, due to it is very unstable. We have also tried to detect the **int-3** by GC-MS analysis but still did not observe. Therefore, we conclude that retro-Benzilic acid rearrangement of **int-3** was occurred instantaneously in the absence of Rh₂(OAc)₄.

(4) **Comments:** “3. Scheme 2, in the graph, scope A, the Ar should be R3? It is very

confusing.”

Our reply: Thanks for this good advice, and proper correction has been made in the revised version. (see: Fig. 3)

(5) **Comments:** “4. Page 4, line 7, left column, “4ÅMS” a space was missing.”

Our reply: Thanks for your advice. The changes have been made in the revised version.

(6) **Comments:** “5. Scheme 4c, the control experiment on reaction between **1a** and **2x** was carried out, but no diketone product was observed. I wondered what products they have detected, any [2,3]-sigmatropic rearrangement product? or O-H insertion product? or something else? Maybe this information could help the author further understand the mechanism of the current reaction.”

Our reply:

Many thanks for pointing out this important issue. Indeed, only the O-H insertion product was obtained (68% yield). This result probably supported that π - σ (C-OH)- π structure of alcohol is essential to the sequential ylide formation/[2,3]-sigmatropic rearrangement and reverse benzylic acid rearrangement to form the 1,2-diketone. The corresponding data have attached in revised MS (Fig. 5) and SI.

(7) **Comments:** “6. Scheme 4f, they found both SiO₂ and the chiral thiourea Q9 could accelerate the isomerization of alkynes. The base could promote this process is reasonable, since MgO they used in the reaction worked well, Q9 bearing two tertiary amine functional groups could play as the base. How to explain the SiO₂ works as well?”

Our reply: Thank you for pointing out this important issue. According to the related references (Hoveyda, et al, *Angew. Chem. Int. Ed.* **2013**, *52*, 7694–7699; Ruano et al. *Chem. Eur. J.* **2012**, *18*, 9775–9779; Kanai, et al. *Chem* **2019**, *5*, 585–599.) and the new experiments results (See: **Scheme 4g**, revised version), the proposed mechanistic

model for the retained enantioselectivity in the alkyne-to-allene isomerization process under Lewis acidic SiO₂ and thiourea catalyst as shown as follows:

a) Base(Q₉) model:

b) SiO₂ model:

(8) Comments: “7. I suggest moving the scheme 5a to the SI file, since this transformation may weaken the theme.”

Our reply: Thanks for your good suggestion, correction has been made in revised version (see: Fig. 6 and SI).

(9) Comments: “8. Can they make a gram scale synthesis to illustrate the reaction practicability?”

Our reply:

We are grateful for this reviewers' comments. As suggested, a gram scale synthesis was conducted, and the results have been added to the revised (Fig. 6a).

(10) **Comments:** “9. The “Discussion” section should be conclusion.”

Our reply: Thanks for this suggestion. We changed the original “**Results and Discussion**” section into “**Results**”, and kept the “**Discussion**” section in the revised manuscript. “**Conclusion**” section is not allowed in “formatting instructions of *Nature Communication*”.

(11) **Comments:** “10. The format of the references are not unified.”

Our reply: Thank you for your kind advice. The correction has been made in revised version.

Responds to the reviewer #3's comments:

(1) **Overall Comments:** “This manuscript describes the discovery of enyne diketones as new substrates for Nazarov cyclization to construct chiral cyclopentenones bearing allene functionality. The reaction involves multi-step processes including rhodium-catalyzed tandem oxonium ylide formation/[2,3]-sigmatropic rearrangement, reverse benzylic acid rearrangement, organo-catalyzed Nazarov cyclization, and alkyne-to-allene isomerization. The results are interesting. **[Comments A]:** “However, the key chemistry of rhodium-catalyzed reaction of allylic as well as propargylic alcohols with diazoacetates via tandem oxonium ylide formation followed by [2,3]-sigmatropic rearrangement (Davies, 2010, 2012), and the Nazarov cyclization of dienyl diketones (Frontier, 2011-2015) has been known. The new aspect is that enyne diketone intermediates obtained by rhodium catalysis was also found to undergo a similar Nazarov cyclization. Thus, I think the conceptual novelty of this manuscript is still limited due to the work by Davies and Frontier.” **[Comments B]:** “Moreover, mechanistic novelty of stereochemical induction is also an important factor. There is no discussion on what is the rational of how the thiourea catalysts helped to control the stereoselectivity and what is the mechanistic insight for stereospecific alkyne-to-allene isomerization.” **[Comments C]:** “On the other hand, this manuscript is not well-organized and English has many errors and grammatical problems that need improvement. Overall, in this reviewer’s opinion, this work is likely more suitable for publication in a less competitive journal such as Chem. Sci. after proper alteration.”

Our reply to the [comment A]:

a) Substrate Design:

b) Davies' works (only **alcohol products** were obtained)

c) Our strategy

Scheme 4a-c:

We are grateful for the reviewers' comments about the novelty of our manuscript.

There is no doubt that Prof. Davies is a giant in metal-carbene chemistry, and we have learn a lot from his contributions. Prof. Davies have reported many elegant works on Rh-catalyzed rearrangement reactions of allylic and propargylic alcohols with diazoacetates. In those rearrangement reactions, a structural characteristic should be noted for allylic and propargylic alcohols w: $\pi\text{-}\sigma(\text{C-OH})\text{-}\sigma$; and alcohol products were obtained in their works. In this work, a new structural characteristic: $\pi\text{-}\sigma(\text{C-OH})\text{-}\pi$ was employed, and an unexpected result was observed. These results probably suggested that the $\pi\text{-}\sigma(\text{C-OH})\text{-}\pi$ moiety is essential structure of alcohol to initiate the sequential ylide formation/[2,3]-sigmatropic rearrangement and reverse benzylic acid rearrangement, to form the 1,2-diketone. Therefore, our reaction is different with Davies' work, especially in terms of novel substrates and novel structure.

a) **Diényl diketones:** developed by Frontier (only 7 examples shown in asymmetric version)

b) **Enyne diketones:** A new substrate classes in asymmetric Nazarov cyclization (**This work**)

Compared with Frontier's diényl diketones and our new enyne diketones, we find that:

Diényl diketones:

a) $\pi\text{-}\pi\text{-}\pi\text{-}\pi$ conjugated system; b) the requirement of additional external **NuH** to initiate the electrocyclozation; c) failed to generate complex chiral skeleton.

Enyne diketones:

a) $\pi\text{-}\pi\text{-}\sigma\text{-}\pi\text{-}\pi$ conjugated interrupted system; b) tandem asymmetric electrocyclozation and proton migration process; c) providing chiral allene cyclopentenone containing a tetrasubstituted carbon atom with central-chirality and an axially chiral allene moiety. The difference of these two diketones not only in mechanism but also in the products. Based on these points, we believe that this novelty of our work is suitable for publication in Nature communications.

Our reply to the [comment B]:

Thanks for the reviewer's comments and we fully understand the expert's concerns on the feasibility of the proposed mechanistic insight. First, some changes have been made in the revised main text **Fig. 2e**. In addition, the mechanism of thiourea catalyst

helped to control the stereoselectivity has been reported in many reported literatures, for example, see the references 30 (Tius *et al. J. Am. Chem. Soc.* **2010**, *132*, 8266–8267).

a) Hoveyda's work (ACIE, 2013)

b) Kanai, enantioselective proton migration for allene synthesis (Chem, 2019)

c) Proposed mechanism of stereospecific alkyne-to-allene isomerization of this work

stereospecific 1,3-proton transfers

On the other hand, the mechanistic insight for stereospecific alkyne-to-allene isomerization of our work is also clear, as shown in above scheme. The mechanisms refer to the reported literature (Hoveyda, *et al. Angew. Chem. Int. Ed.* **2013**, *52*, 7694–7699; Ruano *et al. Chem. Eur. J.* **2012**, *18*, 9775–9779; Kanai, *et al. Chem* **2019**, *5*, 585–599.). And these references have been added in the revised version.

Our reply to the [comment C]:

Thanks for your comments and I'd like to apologize for our carelessness, the errors and grammatical problems has been corrected in the revised version.

(4) Comments: “1. What does the dr refer to? Please clearly indicate the structure of the minor isomer. The details for determination of the dr ratio by crude NMR should be included in the Supplementary Material (SI).”

Our reply:

Thanks for your comments. The desired products contain a central-chirality (*R/S*) and an axially chirality (*R/S*). Therefore, theoretically, there are four isomers: (*R,R*)-**4**, (*R,S*)-**4**, (*S,R*)-**4** and (*S,S*)-**4**. The dr here refer to the ratio of [(*R,R*)-**4** + (*S,S*)-**4**] : [(*R,S*)-**4** + (*S,R*)-**4**]. The enantiomers of (*R,R*)-**4** / (*S,S*)-**4** or (*R,S*)-**4** / (*S,R*)-**4**, have the same *R_f* value and same NMR spectrum. And the diastereomers [(*R,R*)-**4** or (*S,S*)-**4**] and [(*R,S*)-**4** or (*S,R*)-**4**] have different NMR data, usually with different *R_f* value on TLC. In fact, during our experiment, only one single product with single NMR data was obtained. Thus, no minor isomer obtained since the excellent diastereoselective control. See the TLC model above.

(5) Comments: “2. Please clearly indicate whether the formation of alkyne-containing alcohol product was observed. It would be of great interest if both alkyne-containing and allene-containing cyclopentenones could be obtained under proper conditions.”

Our reply: This is a very good suggestion. Indeed, the alkyne-containing alcohol (**Int 5**) has not been detected owing to they are very unstable. Although the intermediates (**Int 5**) of some specific substrates could be detected by TLC analysis during investigating the substrates scope, it cannot be isolated. In order to further explain the

mechanism, many attempts have been made. Finally, only the low reactivity alkyne-containing alcohol **36'** was isolated (see: main text, Fig. 5e). There should be noted that **36'** could also isomerization to allene without any catalyst in solvent. Therefore, in our opinion, it's difficult to selectively obtain the alkyne-containing alcohol product.

(6) Comments: “3. The preparation and characterization data of thiourea catalysts should be provided in SI.”

Our reply: Thanks for your advice. Indeed, all thiourea catalysts used in this paper are known compounds and commercially available. As noted by this reviewer, we provided the preparation and characterization data **Q9** in the revised Supporting Information.

(7) Comments: “4. In most cases, 1,4-enyne alcohols with R1 as ethyl and R2 as methyl were employed.”

Our reply: We appreciate this comment. At the beginning of the project, many attempts have been performed. The results revealed that the R¹ could be H and alkyl groups, and R² must be alkyl. The best result was observed when R² = Me. Also, 2-methyl-2-pentenal is cheaper.

(8) Comments: “5. It seems not clear why the addition of D₂O gave D-4 with only 30% deuterium labeling (Scheme 4d).”

Our reply: Thank you for your question. I read a lot of related reported literatures. We believe that this reaction was through a “1,3-Proton Migration” pathway:

As a result, there is a competition between a hydrogen atom and a deuterium atom. And new additional experiments were performed these days. It is interesting that when the **Q9** instead of MgO, the D-4 was increased from 30% to 70% deuterium labeling. The ¹H-NMR spectrum as shown as follows:

(9) Comments: “6. What is the mechanistic model for the retained enantioselectivity in the alkyne-to-allene isomerization process under Lewis acidic SiO_2 and thiourea catalyst (Scheme 4f)?”

Our reply:

a) Base(Q9) model:

stereospecific 1,3-proton transfers

b) SiO₂ model:

stereospecific 1,3-proton transfers

We appreciate this comment. According to the related references (Hoveyda, et al, *Angew. Chem. Int. Ed.* **2013**, *52*, 7694–7699; Ruano et al. *Chem. Eur. J.* **2012**, *18*, 9775–9779; Kanai, et al. *Chem* **2019**, *5*, 585–599.) and the new experiments results (See: **Fig. 5g**, revised main text), the proposed mechanistic model for the retained enantioselectivity in the alkyne-to-allene isomerization process under Lewis acidic SiO₂ and thiourea catalyst as shown as above.

(10) Comments: “7. Please provide a reaction pathway to illustrate the formation of product 53 in Scheme 5c.”

Our reply: We appreciate the suggestion, the proposed pathway has been added in revised SI, page. Also list as follows:

proposed pathway:

REVIEWER COMMENTS

Reviewer #2 (Remarks to the Author):

The revised version of this manuscript has addressed all the issues on my previous questions and suggestions. I don't have additional comments.

Reviewer #3 (Remarks to the Author):

The authors revised the manuscript and made their efforts to respond to the raised points. I would recommend its publication after some further revision.

- 1) About dr, if no minor isomer formation was observed, please indicate in the main text. It is confusing to give a ratio of >19:1, as this is likely unusual.
- 2) For the deuterium labeling experiments (Fig 5d), a new result with largely improved ratio (70% D) under the conditions of MgO/D₂O was obtained, please provide an explanation in the main text.
- 3) In the enantioselective version (Table 3), products 27 and 44 are missing (they are listed in Fig 3).
- 4) Although the retained enantioselectivity in the alkyne-to-allene isomerization process (deprotonation/stereospecific protonation) is not clear, the proposed mechanistic models under Lewis acidic SiO₂ and thiourea catalyst are suggested to be included in the SI.
- 5) The structure of product 53 in the proposed pathway in SI is wrong (Page 26). Please also check the experimental description, the starting compound is NOT 3aq, and the product should NOT be 7. Similarly, check Page 29.
- 6) Following the proposed pathway for the formation of product 53, the claim of a dr ratio of >19:1 is very confusing as a racemic product should be obtained. Please provide clear supports (HPLC, ee, optical rotation) to confirm the stereochemical result, and also indicate it in the main text.

Dear reviewers,

Thank you very much for your suggestions. We have carefully revised this manuscript again based on your valuable comments. The corrections are given in the revised manuscript in detail and have been highlighted in yellow. The detailed revisions described below, in the responses to the reviewers' comments:

Reviewer #2 (Remarks to the Author):

“The revised version of this manuscript has addressed all the issues on my previous questions and suggestions. I don't have additional comments.”

Our reply: We thank the reviewer for taking the time and effort to review our manuscript and for providing positive comments and suggestions of our work.

Reviewer #3 (Remarks to the Author):

“The authors revised the manuscript and made their efforts to respond to the raised points. I would recommend its publication after some further revision.”

Our reply: Many thanks for these positive comments. Your kind suggestions also help us a great in further improving our manuscript.

(1) **Comments:** “About dr, if no minor isomer formation was observed, please indicate in the main text. It is confusing to give a ratio of >19:1, as this is likely unusual.”

Our reply: Thank you for your kind advice. The dr has indicated in the revised main text.

(2) **Comments:** “For the deuterium labeling experiments (Fig 5d), a new result with largely improved ratio (70% D) under the conditions of MgO/D₂O was obtained, please provide an explanation in the main text”.

Our reply: Thank you for your good advice. Owing to the competition between a

hydrogen atom and a deuterium atom, and the influence of catalyst (MgO or Q9), the results were that when the Q9 instead of MgO, the D-4 was increased from 30% to 70% deuterium labeling. The explanation has been provided in revised text.

(3) **Comments:** “In the enantioselective version (Table 3), products 27 and 44 are missing (they are listed in Fig 3)”.

Our reply: Thank you for your good question. Although the products 27 and 44 were prepared successfully in the racemic conditions (Method A: [Rh]/MgO, Fig 3), they were not obtained in the asymmetric conditions (Method B, [Rh]/Q9). The probably reason was that the MgO exhibit higher catalytic activity than Q9. The table 3 has been updated, and also noted in the revised main text (page 8, left column).

(4) **Comments:** “Although the retained enantioselectivity in the alkyne-to-allene isomerization process (deprotonation/stereospecific protonation) is not clear, the proposed mechanistic models under Lewis acidic SiO₂ and thiourea catalyst are suggested to be included in the SI.”

Our reply: Thank you for your good suggestion. The correction has been made in revised SI (page 9).

(5) **Comments:** “The structure of product 53 in the proposed pathway in SI is wrong (Page 26). Please also check the experimental description, the starting compound is NOT 3aq, and the product should NOT be 7. Similarly, check Page 29.”

Our reply: We thank the reviewer for mentioning the errors. The correction has been made accordingly (page 27).

(6) **Comments:** “Following the proposed pathway for the formation of product 53, the claim of a dr ratio of >19:1 is very confusing as a racemic product should be obtained. Please provide clear supports (HPLC, ee, optical rotation) to confirm the stereochemical result, and also indicate it in the main text.”

Racemization process

proposed pathway:

Our reply: We thanks the reviewer's question. The product 53 certainly as a racemic product, we can see from the above pathway. It's a racemization process from **20** to **int B**. Dr ratio mean that this reaction has high diastereoselectivity, but not enantioselective retention. The result product 53, $[\alpha]_D^{20} = 0$ (0.5, CHCl_3) and ee = 0%,

also support our proposed mechanism.

HPLC of 53

Peak Name	RT [min]	Type	width [min]	Area [mAU*s]	Height [mAu]	Area ratio %
1	7.941	MM	0.2245	3.56464e4	2646.50000	49.2785
2	9.064	MM	0.2371	3.66903e4	2579.48120	50.7215

REVIEWERS' COMMENTS

Reviewer #3 (Remarks to the Author):

It seems that the authors have addressed most of my points. I don't have further comments.